# Feasibility Trial of Yoga Programme for Type 2 Diabetes Prevention (YOGA-DP) among High-Risk People in India: A Qualitative Study to Explore Participants’ Trial- and Intervention-Related Barriers and Facilitators

**DOI:** 10.3390/ijerph19095514

**Published:** 2022-05-01

**Authors:** Pallavi Mishra, Tess Harris, Sheila Margaret Greenfield, Mark Hamer, Sarah Anne Lewis, Kavita Singh, Rukamani Nair, Somnath Mukherjee, Nandi Krishnamurthy Manjunath, Nikhil Tandon, Sanjay Kinra, Dorairaj Prabhakaran, Kaushik Chattopadhyay

**Affiliations:** 1Centre for Chronic Disease Control, New Delhi 110016, India; pallavi@ccdcindia.org (P.M.); kavita@ccdcindia.org (K.S.); dprabhakaran@ccdcindia.org (D.P.); 2Population Health Research Institute, St. George’s University of London, London SW17 0RE, UK; tharris@sgul.ac.uk; 3Institute of Applied Health Research, University of Birmingham, Birmingham B15 2TT, UK; s.m.greenfield@bham.ac.uk; 4Institute Sport Exercise and Health, Division of Surgery and Interventional Science, University College London, London W1T 7HA, UK; m.hamer@ucl.ac.uk; 5Lifespan and Population Health Academic Unit, School of Medicine, University of Nottingham, Nottingham NG5 1PB, UK; sarah.lewis@nottingham.ac.uk; 6Bapu Nature Cure Hospital and Yogashram, New Delhi 110091, India; research@bnchy.org (R.N.); soumikmukherjee987@gmail.com (S.M.); 7Swami Vivekananda Yoga Anusandhana Samsthana, Bengaluru 560105, India; nkmsharma@svyasa.org; 8All India Institute of Medical Sciences, New Delhi 110029, India; nikhil_tandon@hotmail.com; 9Department of Non-Communicable Disease Epidemiology, London School of Hygiene and Tropical Medicine, London WC1E 7HT, UK; sanjay.kinra@lshtm.ac.uk

**Keywords:** Yoga, physical activity, barriers, facilitators, prevention, type 2 diabetes, prediabetes, qualitative research, randomised controlled trial, feasibility

## Abstract

Yoga-based interventions can be effective in preventing type 2 diabetes mellitus (T2DM). We developed a Yoga programme for T2DM prevention (YOGA-DP) and conducted a feasibility randomised controlled trial (RCT) among high-risk people in India. This qualitative study’s objective was to identify and explore participants’ trial- and intervention-related barriers and facilitators. The feasibility trial was conducted at two Yoga centres in New Delhi and Bengaluru, India. In this qualitative study, 25 trial participants (13 intervention group, 12 control group) were recruited for semi-structured interviews. Data were analysed using deductive logic and an interpretative phenomenological approach. Amongst intervention and control participants, key barriers to trial participation were inadequate information about recruitment and randomisation processes and the negative influence of non-participants. Free blood tests to aid T2DM prevention, site staff’s friendly behaviour and friends’ positive influence facilitated trial participation. Amongst intervention participants, readability and understanding of the programme booklets, dislike of the Yoga diary, poor quality Yoga mats, difficulty in using the programme video, household commitment during home sessions, unplanned travel, difficulty in practising Yoga poses, hesitation in attending programme sessions with the YOGA-DP instructor of the opposite sex and mixed-sex group programme sessions were key barriers to intervention participation. Adequate information was provided on T2DM prevention and self-care, good venue and other support provided for programme sessions, YOGA-DP instructors’ positive behaviour and improvements in physical and mental well-being facilitated intervention participation. In conclusion, we identified and explored participants’ trial- and intervention-related barriers and facilitators. We identified an almost equal number of barriers (*n* = 12) and facilitators (*n* = 13); however, intervention-related barriers and facilitators were greater than for participating in the trial. These findings will inform the design of the planned definitive RCT design and intervention and can also be used to design other Yoga interventions and RCTs.

## 1. Introduction

The 2019 International Diabetes Federation report estimated that 77 million people in India were living with diabetes, with this number expected to increase to 101 million by 2030 [1]. Another 77 million people in the country are at high risk of developing type 2 diabetes mellitus (T2DM) because of raised blood glucose levels but below the established threshold for T2DM [2]. One of the major contributors to T2DM is an unhealthy lifestyle, including physical inactivity and an unhealthy diet [3]. Past studies have found lower physical activity levels and unhealthy dietary practices among people living in India [2,4,5]. Therefore, screening people at high risk of developing T2DM and subsequently offering them an effective lifestyle intervention could be a cost-effective prevention strategy [3].

Evidence suggests that health interventions are more successful if informed by people’s local socio-cultural expectations and health beliefs [6]. Yoga has traditionally been part of the Indian culture, and Yoga-based interventions are more likely to be accepted by Indians [7,8]. Yoga incorporates physical activities and a healthy diet, which results in a disciplined body and mind [9]. Yoga-based interventions can be easily replicated across diverse populations and settings. Yoga relies on a gentle approach, and these interventions can be delivered with a low to moderate level of guidance [7]. Other benefits of a Yoga-based intervention are that it is less expensive to run and can also be practised at home or indoors [7]. Yoga is considered to be safe and can be practised by people with a range of comorbidities as it comprises low- and moderate-intensity activities that help strengthen muscles [7,10]. Previous studies have generated some evidence on the beneficial effects of Yoga in T2DM and metabolic syndrome [11,12,13,14], suggesting that Yoga has the potential to prevent T2DM among high-risk individuals.

We developed a Yoga programme for T2DM prevention (YOGA-DP) among high-risk populations in India. The YOGA-DP is a structured lifestyle education and exercise intervention provided over 24 weeks [15]. A randomised control trial (RCT) is planned to evaluate its effectiveness. Prior to this definitive RCT, we conducted a feasibility RCT [16], as part of which we used semi-structured interviews to identify and explore participants’ trial- and intervention-related barriers and facilitators. Additionally, we conducted semi-structured interviews with individuals who declined to participate in the study, published elsewhere [17].

## 2. Materials and Methods

The detailed feasibility RCT protocol is published elsewhere [15]. The feasibility trial was carried out at two Yoga centres—one in northern India (Bapu Nature Cure Hospital and Yogashram (BNCHY, New Delhi, India)) and one in southern India (Swami Vivekananda Yoga Anusandhana Samsthana (SVYASA, Bengaluru, India)). We used a multipronged approach, such as door-to-door campaigns, posters and screening camps to identify potential participants, i.e., adults at high risk of T2DM as their fasting blood glucose (FBG) level was between 100 and 125 mg/dL [18].

The intervention: YOGA-DP intervention comprises of structured lifestyle education and a Yoga-based exercise programme, which was provided over 24 weeks. This programme was delivered by qualified YOGA-DP instructors (male and female), who were also provided with formal training on the intervention. These YOGA-DP instructors ran the group Yoga sessions locally (such as at Yoga centres and community centres) at different times (with evening and weekend sessions) so that participants could attend the Yoga session at their convenience. During the first three months, the participants were delivered at least two YOGA-DP sessions weekly at the session site, and they were advised to follow YOGA-DP booklet part I (giving information on their condition (i.e., at ‘high risk’ of developing T2DM) and on how to prevent T2DM). In the remaining three months, they were provided with a YOGA-DP booklet part II (giving information on yoga practice to prevent T2DM) and a YOGA-DP video on a USB flash drive to practice Yoga at home without any supervision, and they scheduled at least one session every four weeks at the session site [15]. The YOGA-DP instructor also phoned them weekly to offer support and help and to resolve any problems with their unsupervised sessions. For the unsupervised sessions, participants were also provided with a YOGA-DP diary for self-reporting of Yoga practice, including types and minutes [15].

The control group: in India, there is no standard lifestyle intervention for people at high risk of T2DM, thus, control group participants received a leaflet on routine lifestyle advice to prevent T2DM [15]. The leaflet was provided by a different team member (i.e., not by the YOGA-DP instructor) to avoid contamination. This might have helped in lowering the attrition and made the control group participants feel that there were benefits to participation [15].

For this qualitative study, after the completion of the feasibility trial, intervention and control group participants who had completed six months of participation were recruited from both trial sites. One intervention group participant who stopped the intervention but continued participating in the trial was also interviewed to understand the barriers which hindered participation in the intervention group. They were invited to participate in a semi-structured interview with a trained qualitative researcher from the Centre for Chronic Disease Control (CCDC), New Delhi, India, not involved in the feasibility trial participant recruitment. Out of the total 65 participants who were randomised in the feasibility trial, 25 (38%) (13 intervention group, 12 control group) agreed to participate in the semi-structured interview. These interviews were conducted to identify and explore participants’ trial and intervention related barriers and facilitators. Although we started reaching data saturation by the sixth and seventh interviews in each group, we conducted further interviews to ensure that we did not miss any new and unique information and to cover a broader socio-demographic spread of participants [19,20].

The participant information sheet and consent form (available in Hindi, Kannada and English) were read and shared with participants before the interview. Written informed consent was obtained from those interested in participating, including for digital recording of interviews. The qualitative researcher conducted interviews, either face-to-face at the participants’ house (*n* = 12) or by telephone (*n* = 13) [21,22], depending on availability and convenience, from December 2019 to July 2020. The researcher spoke to the participants two to three times over the phone before the actual interview to become acquainted and make them comfortable about the interview process. Interviews were conducted in Hindi (*n* = 22), Kannada (*n* = 2) or English (*n* = 1) as preferred by the participants and using a pre-tested interview guide available in these languages. The female qualitative researcher is bilingual and fluent in English and Hindi and has experience in non-communicable diseases and health systems research in low- and middle-income countries. Kannada interviews were conducted with the help of an interpreter, who translated the interview questions from English to Kannada for participants and participants’ responses from Kannada to English for the researcher. The interpreter supported the researcher in the process of in-depth probing. The interview guide had some key open-ended questions, which were adapted to the participants’ responses and probed further to capture detail and develop a deeper understanding of each answer. The interviews were digitally recorded and transcribed verbatim by professional transcribers and then translated from Hindi to English by a professional translator if needed. Kannada interviews were transcribed and translated into English by the interpreter hired to assist in conducting Kannada interviews. The qualitative researcher ensured the quality of the final transcripts (for interviews in Hindi and English) by listening to the audio files and constantly comparing these with the transcripts to rule out the possibility of any missing data by the transcriber. Similarly, to ensure the quality of the Kannada transcripts, a local study staff member at SVYASA listened to the audio files in Kannada and compared these with the transcript to rule out the possibility of any missing data by the transcriber.

During and after data collection, the qualitative researcher familiarised herself with the data by reading the transcripts multiple times. The researcher coded all the interviews using the interpretative phenomenological approach (IPA), and deductive logic with the help of QSR-NVivo 10 software for data management [23,24]. IPA helped in capturing the detailed overview of participants’ lived experiences, as it is not bound by any pre-existing theoretical preconceptions [23]. Interviews with the participants from both groups helped identify and explore the trial-specific barriers and facilitators. The interviews with intervention group participants were useful in understanding the specific barriers and facilitators of participation in the intervention. The researcher coded and analysed the interviews of the control group first and then the intervention group.

The original data were reflected continuously to ensure participants’ experiences and perceptions were accurately captured. In the first stage of coding, the response from the participants was coded using meaningful chunky statements, and in the second stage, the chunky statements were reduced to fewer words to move closer to the ‘core essence’ [25]. Finally, in the third stage, the researcher encapsulated the ‘core essence’ of the central meaning of the participants’ lived experiences in one or two words [25]. After assigning codes to transcripts, summaries were prepared from the coded data. The summaries were organised into overarching categories and later assigned themes and sub-themes. The process was continuously discussed with the study investigators (including a senior qualitative researcher) for refinement. Once the emergent themes were captured, the researcher identified the connection between themes and organised them in chronological order, the way they emerged in the transcript and these themes were further organised in theoretical order to make sense of the connection between themes [25]. The data were interpreted using the language, metaphors, symbols, repetitions, pause and context of the participants and the initial reflexivity of the researcher using field diary and reflexivity notes [26]. The researcher was aware of her age, gender, affiliation with the study site (not a BNCHY staff member), the place of the interviews while preparing the reflexivity notes and their impact on interviews and interactions with participants.

Ethics approval for this study was obtained from the following research ethics committees: Faculty of Medicine and Health Sciences, University of Nottingham (UK), CCDC (India), BNCHY (India) and SVYASA (India).

## 3. Results

The median age of participants was 41 years, and 13 participants (out of 25) were female (Table 1). Interview duration ranged from 20 to 60 min (average 37 min). The average of the telephonic interviews was 33 min, and for face-to-face interviews, it was 41. Similarly, the average duration of Hindi interviews was 35, and for Kannada, it was 49 min. All the participants were married, and four of them had less than ten years of formal education. The gross monthly income of the participants ranged between INR 10,000 and 245,000. There were 11 participants who had a family history of diabetes. We identified and explored participants’ trial- and intervention-related barriers and facilitators. The findings are presented through four major themes, each of which has several sub-themes (Table 2). All participant quotes represented below are anonymised but attributed by age, gender and study arm.

### 3.1. Barriers to Trial Participation

This theme captures the potential barriers that influenced a participant’s decision to participate in the feasibility trial. This theme captures the experiences and perceptions of both control and intervention group participants.

**Detailed information about recruitment and randomisation processes**: Some participants were either unaware of the recruitment and randomisation processes or did not have adequate information about their recruitment to a particular group. Some of them thought they were recruited to a specific group because of their age and blood test report.
*“As far as I know, my sugar level (diabetes) is on the borderline, that is why. If I practice Yoga, I can prevent diabetes. They said that your sugar level is on the borderline if you continue practising Yoga, you can control (prevent T2DM)”.*(Age: 33 years; Female; Intervention)

Some participants assumed they were recruited to the control group because they had informed the site staff about their busy schedule, and also, they were not interested in practising Yoga. At the same time, many of them were not aware of the intervention group and mentioned their inclination toward joining the intervention group. Lack of detailed information about recruitment and randomisation and desire to get recruited in a particular group could have been a barrier in retaining participants in the trial.
*“Because I told them that I have lots of work to do in the morning and I do not have interest in it, so they did not keep me in Yoga group”.*(Age: 43 years; Female; Control)

**Poor experience in the control group regarding the enhanced care leaflet**: Some control group participants said they did not receive the enhanced care leaflet meant for the control group from the site staff.
*“No, I did not receive that. I just got a photocopy of my reports”.*(Age: 64 years; Female; Control)

Many participants acknowledged receiving the enhanced care leaflet, but they claimed that it was not of any use to them as they were already aware of the information mentioned in the leaflet. Some of them also misunderstood it only as a diet chart to be followed to prevent T2DM.
*“That (leaflet) had suggestions to increase everything that one was doing. As the Research Assistant asked me to increase exercise a little bit. Have more green vegetables… I got a little help. Nothing much because I knew everything (that was mentioned in the leaflet)”.*(Age: 47 years; Male; Control)

**The negative influence of non-participants**: Some non-participants in this study had tried to dissuade participants from taking part in the feasibility trial as they were of the view that the glucometer used at the time of screening gave false results.
*“Other people also got their tests done and learnt about their high Sugar, so they told us not to pay much attention and said that their machines were not working properly. There is no need for it; nothing will happen, and that it is not worth it”.*(Age 44 years; Female; Intervention)

**Frequency of the blood test (e.g., FBG)**: The majority of participants opined that the blood test should be conducted once in the middle of the feasibility trial. They believed that the blood test after three months could have given a report about their health, and they could have taken health-related decisions based on that.
*“After receiving my recent report, I felt that if the test was done after three months, then I would have known that my sugar level had increased from 102 to 105 in 3 months, then I would have been more serious”.*(Age: 27 years; Female; Control)

### 3.2. Facilitators to Trial Participation

This captures the factors that facilitated the participation and retention of participants in the intervention and control group.

**Adequate information about the feasibility trial and processes**: Some participants were employed at the site and were informed about the blood test and the YOGA-DP intervention by site staff. They were also aware that they were recruited into the feasibility trial as they were at risk of developing T2DM. These factors helped participants make an informed decision about taking part and adhering to the trial processes.
*“They informed me that Yoga sessions would take place, and I shall have to participate in that. They told me that they were doing some research, and they would assess the report of my blood test and decide. They told me that I would be given Yoga sessions at the site, and after that, I shall have to practice that at home”.*(Age: 43 years; Male; Intervention)

Some of the participants shared that they understood the randomisation and recruitment processes. They mentioned that the site staff had informed them about randomisation and recruitment to their respective groups, and they decided to be a part of that group and follow the processes.
*“They told me that one would be the Yoga group, and in the other group, only blood test and routine check-ups would be done. They told me that they would decide which group I would be recruited to. So, I agreed to that. So, they just gave me information that two groups will be formed”.*(Age: 34 years; Female; Intervention)

**Free blood tests, positive experiences of the testing process**: Many of the control group participants confirmed that they came for the screening because of the free of cost capillary blood test and confirmatory venous blood test.
*“I have to attend every camp which is organised free of cost… Now I usually have to go to the hospital for a check-up after making an appointment. After scheduling an appointment, I have to fast. Then they will check, and then the report comes on the next day; it becomes a bit hectic. That is why I do not prefer it”.*(Age: 47 years; male; Intervention)

None of the participants reported facing challenges during the blood sample collection, which was a positive experience.
*“It was good, and there was no problem”.*(Age: 43 years; Female; Control)
*“No, I did not face any problem”.*(Age: 43 years; Male; Intervention)

Some participants mentioned that they appeared for the blood test only because the behaviour of the site staff made them comfortable, and they decided to appear for the test. Some of them also shared that the appointment for the blood test was made as per the participant’s preference, because of which they did not face any problem during the blood test.
*“… the person who collected my blood sample was very cooperative. When I went for the blood test he gave me 15–20 min of his time after every other patient left. Even last time, I was allotted 10:30 AM, but it took 15 min extra for me as I kept pulling away my arms out of fear. But he was very supportive”.*(Age: 27 years; Female; Control)

Some participants also mentioned that they trusted the laboratory and procedures followed during the blood test. Trust in the blood collection process and testing laboratory also positively influenced the participant’s decision to come for the blood test.
*“…there was nothing lacking as the laboratory where they performed the tests is a good one. I observed their process and (experienced) how responsibly they were handling us and performing our tests, which was very good to see”.*(Age: 39 years; Female; Control)

Some of the control group participants went for the blood test to find out if their glucose levels had increased. They mentioned that usually they went to a nearby dispensary to have a diabetes blood test but decided to come to the feasibility trial site as the test was already happening here
*“I thought that at least I would get to know if I have Sugar or not. I get it (test) done sometimes at the dispensary also. I decided to get it done since they were doing it. I didn’t see any problem in doing so. I get it (test) done sometimes at the dispensary also…So I went for the tests straight away as I had not eaten since morning”.*(Age: 64 years; Female; Control)

**To gain adequate information to prevent T2DM**: The motivation of the control group participants to participate in the feasibility trial was to prevent T2DM. They found it easier by enrolling in a programme that focused on T2DM prevention.
*“The biggest motivating to participate in this study was it was a win-win opportunity for me because I had to control (prevent) diabetes anyway. And you also wanted to recruit a person who is willing to control his diabetes. So, there was nothing to lose. So, I was motivated, and I thought that this was such good research and I should also become a part of it”.*(Age: 47 years; Male; Control)

**Professional behaviour of the site staff**: While talking about their experience at the trial site, many participants mentioned that the site staff were friendly. The behaviour of the site staff influenced many control group participants and positively impacted their decision to participate in the feasibility trial.
*“The best part I felt was your team; they supported me completely. They managed time according to my availability, whether it was the Trial Coordinator or the Research Assistant. So, it was nice. Instead of dictating us on time, you changed your timing according to the participant or client. Even if there was a delay, you adjusted accordingly. So, you gave priority to our preferences”.*(Age: 47 years; Male; Control)

**The positive influence of friends**: Many participants said their friends suggested that they go for the blood test. Friends also played a vital role in motivating participants to go for the blood test and participate in the feasibility trial.
*“They called everybody in my neighbourhood saying that they were conducting free blood tests, and everybody asked me, “Why do not you come?” So, I went for the tests as I was already on an empty stomach at that time”.*(Age: 43 years; Female; Control)

**Trust in the trial sites and the range of healthcare services they provide**: Some other participants trusted the health facility (feasibility trial site) and its staff; thus, they decided to come for the blood test. Trust was one of the key factors which facilitated participation in the intervention group.
*“I went there because a camp was organised. I knew about Bapu Nature (trial site) for quite some time. When I came here for tests, they told me that my sugar level was high”.*(Male; Age: 44; Intervention)

### 3.3. Barriers to Participation in the Intervention

This theme captures the barriers shared by the participants from the intervention group, which may have a bearing on adherence to the programme sessions.

**Difficulty in reading and understanding the language of the programme booklets**: Some participants mentioned that the readability and understanding of the booklet were difficult, and they were unable to grasp the information provided in them. Because of the difficulty in readability and understanding, some of them could not use the booklet to change their lifestyle and practise Yoga.
*“It should have been easy so that people could understand it easily, like what all to avoid. What habit to develop and what to forgo. So, an easy language will make people understand it more easily”.*(Age: 44 years; Male; Intervention)

**Difficulty in capturing the duration and sequence of the Yoga poses (asanas) in the programme diary**: Participants shared their experience of filling up the programme diary as they could not capture the time spent for Yoga poses. Some of them could fill in the diary and follow the instructions given at the site-based sessions for the same. However, it was difficult for many others to fill in the details in the programme diary.
*“I used to find it difficult early on, but as I started to follow it, I did not find it difficult later. In the initial 1–2 pages, I thought as to how to mention time in this, but I got an idea later on”.*(Age: 35 years; Female; Intervention)*“Yes, it would have been better even without it (diary). If someone does not want to do it, he/she will fill it up without practising Yoga”.*(Age: 44 years; Male; Intervention)

Some participants also mentioned that they did not receive any diary from the site staff. Other participants did not have an adequate understanding of the diary.
*“I have not got the dairy. I follow the books which is given to me and during free time I practice (Yoga) also”.*(Age: 33 years; Female; Intervention)
*“Yes, they asked us to mention what time we got up, what kind of breakfast we had, and what type of food we ate. We also had to mention the time of the food and what time we went to bed (to sleep). We also had to mention the changes that we had witnessed in the past six months”.*(Age: 33 years; Male; Intervention)

Lack of interest in filling in the diary and clarity about the same hindered monitoring of adherence to the unsupervised programme sessions among participants.

**Difficulty in using the programme video during unsupervised sessions**: Intervention group participants were provided with a video to facilitate the unsupervised sessions. Some participants stated that they did not find the programme video useful, and they never used it and some of them used booklets for their unsupervised sessions. Perceived lack of usability of the programme video was a barrier to practising Yoga poses and adhering to the unsupervised sessions.
*“No, I have not watched that video even once, and I have not practiced Yoga since Sir (YOGA-DP instructor) taught us (stopped taking session). I have committed a mistake by not watching that video, but I will definitely watch it”.*(Age: 25 years; Female; Intervention)*“I found it easy with the book. I used to keep the book in front of me and finish the Yoga poses quickly. I was able to do it with the help of the book”.*(Age: 44 years; Female; Intervention)

Some of them suggested providing the web link for the video instead of providing a full-length video in a pen drive. Some of them also mentioned that they could not run the pen drive on their television.
*“If you could give us some link of the website or share any application that would have been helpful as these days people do not use pen drive you could have shared the videos over WhatsApp as well”.*(Age: 33 years; Male; Intervention)

**Poor quality of Yoga mats**: Some participants registered complaints about the hardness of the Yoga mats and noted that the quality of the Yoga mat could have been better. Poor quality Yoga mats caused discomfort among participants and emerged as a barrier in practising Yoga poses.
*“The quality of the mats is not good because it is very thin when you practice Yoga on it. Sometimes it feels that our hand is stuck to the mat. And in the thicker mats, there is no such problem in practising”.*(Age: 47 years; Male; Intervention)

**Household commitment and unavailability of the YOGA-DP instructors hindered unsupervised sessions**: After completing three months of supervised sessions, intervention group participants had to complete 21 unsupervised sessions at home. Most participants said that it was challenging for them to practice Yoga at home due to their household responsibilities. The household commitment was one of the barriers to adherence to the programme sessions.
*“Here, she would make us relax. At home, these things were not possible because we had household responsibilities. I would do the same exercise at home for 35–40 min, but here I might do it for two hours. So, this happened with me at home”.*(Age: 62 years; Female; Intervention)

Some of the participants explained that they could not adhere to the timing and subsequent steps of the Yoga poses in the unsupervised sessions. They also mentioned that there was no one to rectify their mistakes in the unsupervised session and help them with practising Yoga poses. Lack of supervision and support was also a barrier in adherence to programme sessions.
*“Yes, you can say that household responsibility was a factor, but I did not enjoy a Yoga session without supervision. There should be someone to teach us how to do and what to do”.*(Age: 36 years; Male; Intervention)

**Missed supervised sessions due to unplanned travel**: Many intervention group participants had to miss a few sessions due to unplanned travel; however, they attended compensatory sessions after returning. One of the participants also had to rush through one of the sessions because of her unexpected travel.
*“I did not miss it actually, but rushed through my session in a hurry, as there was some family function and I had to leave with someone for that”.*(Age: 35 years; Female; Intervention)*“Sir (YOGA-DP Instructor) used to make me do everything that I used to miss in those two classes. It’s not as if I’ve missed anything”.*(Age: 25 years; Female; Intervention)

**Difficulty in practising Yoga poses**: Some of the intervention group participants said that they did not have any experience of Yoga, and they faced some difficulty in practising different Yoga poses, such as sun salutation (surya namaskar) in the beginning.
*“At the outset, as it happens with everyone, I also had some difficulty, but later on with practice, everything got normal, and I did not face any problem. Everything is a little difficult at the beginning. In the beginning, it was difficult. Later on, there was no problem, and I started liking it (Yoga)”.*(Age: 36 years; Male; Intervention)

They mentioned that with practice, they were able to do it with ease after some time. Some of them could not practice these exercises because of their existing health-related conditions.
*“When I started doing it, sometimes I pulled my veins and developed pain in my legs. When I used to go back after doing Yoga, then while climbing stairs, I felt a pull in my veins. When I continued doing it, it gradually got loose. For a week or two, I used to think, “what I have chosen to be a part of?” I felt as if my body was aching even more. But after continuing with it, it gradually got better”.*(Age: 43 years; Male; Intervention)

Some male participants shared that they had to miss a few sessions because of their health issues, such as fever or hypertension, whereas some female participants missed their sessions on their menstruation days.
*“I missed 1–2 classes due to (increase in) Blood Pressure, but I completed it later on”.*(Age: 43 years; Male; Intervention)
*“I used to miss my classes during my periods, or if I had to go to my mother’s house”.*(Age: 25 years; Female; Intervention)

Difficulty in practising Yoga poses was a barrier for some participants, who had never practised Yoga before. This could have affected adherence to programme sessions among participants.

**Hesitation in attending programme sessions with the YOGA-DP instructor of the opposite sex**: Some of the male participants reported their discomfort while practising Yoga under the supervision of a female YOGA-DP instructor. They mentioned that with male YOGA-DP instructors, they could share their problems openly but not with the female YOGA-DP instructor, which negatively influenced their practice of Yoga poses.
*“…sometimes the Yoga Instructor Sir (YOGA-DP instructor) was not available, so we used to ask the Yoga Instructor Ma’am (YOGA-DP instructor) to do it (to take YOGA-DP sessions), but we used to feel uncomfortable in that. We preferred the Yoga Instructor, Sir”.*(Age: 43 years; Male; Intervention)

Similarly, some female participants shared that they preferred the female YOGA-DP instructor. They felt that they would have felt cautious about their clothes and body movement with a male YOGA-DP instructor. Different gender of the YOGA-DP instructor was also cited as one of the barriers across male and female groups.
*“When I came here, they told me in the first meeting that male participants will get training from the male instructor (YOGA-DP instructor) and females will get from a female instructor (YOGA-DP instructor), which attracted me a lot. Ladies sometimes wear clothes with deep necks, we also have to raise our legs during exercise, everything has to be done, so I liked this thing here that male got training from the male instructor and female got from the female instructor”.*(Age: 44 years; Female; Intervention)

**Hesitation in attending group programme sessions**: Some participants stated that they preferred individual sessions. They felt that in individual sessions, YOGA-DP instructors could pay more attention to one participant, which was not possible in a group session. One of the participants also shared that he was hesitant in attending a group session as he did not want others to laugh at him if he could not do a particular Yoga pose properly. Thus, group sessions were found to be a barrier for some participants in attending the programme sessions.
*“I feel that for those who do not know Yoga for them, individual sessions are better. If I do not know anything, the instructor (YOGA-DP instructor) can pay more attention to teaching me in an individual session than in group sessions. As we were given private sessions, she paid attention to us…”*(Age: 44 years; Female; Intervention)*“People may laugh at you if you practice Yoga in a group. Suppose someone is able to do any Yoga pose, and the other person is not able to do properly, everyone thinks differently, but people may laugh at you”.*(Age: 44 years; Male; Intervention)

Some participants suggested that there should be separate groups for young and old participants as they felt that the pace of doing exercise was slower among older adults. Group composition also influenced the practice of Yoga poses among participants.
*“It should be age-wise. Because Young people are able to do it (practice Yoga) quickly and old people take time in doing it”.*(Age: 40 years; Female; Intervention)

### 3.4. Facilitators to Participate in the Intervention

This theme captures the key factors based on the experiences and perceptions of the participants, which facilitated participation in the intervention group.

**Programme booklets helped in adhering to the unsupervised programme sessions**: While talking about their experience with the programme booklet, some participants felt that they had limited opportunity to read the booklet. They said that they read the booklet as and when they have time. However, many participants also mentioned that they read the booklet and referred to that during the home-based sessions. They used the booklet to know the exact pose and sequence of Yoga poses. They were of the view that pictures were helpful in understanding various steps of the Yoga poses.
*“I used to refer to the book and practice. I used to keep the book with me while practising Yoga at home, it helped me, and otherwise, it was difficult to remember. The book helped me in remembering the steps”.*(Age: 36 years; Male; Intervention)

**Adequate information on T2DM prevention and self-care**: Many participants pointed out that when they got to know that they were at a high risk of developing T2DM, they wanted to prevent the condition, which motivated them to come for the programme sessions.
*“Actually, I had not imagined that I would be pre-diabetic. So, next day only, Sir (Trial Coordinator) told me that you are (at) pre-diabetic (stage) and we are organising Yoga classes, so will you participate in it? So, I replied yes, I will, because I wanted to cure myself, as no one in my family has it (diabetes)”.*(Age: 25 years; Female; Intervention)

Some other participants said that they were motivated to participate in the session because they could take out time for self-care by doing that. Programme sessions gave them a sense of self-care which motivated them to spare time to come for the sessions and practice Yoga.
*“It is just that I was taking out two hours time from my schedule and practising Yoga. The YOGA-DP instructor taught us well… No, I also used to feel relaxed. My body was relaxed, and I felt light”.*(Age: 60 years; Female; Intervention)

Many participants liked being in the intervention group as they learnt something, and also lost body fat. They also mentioned that had they been in the other group, they would not have learnt anything. They believed that they could practice Yoga only because they were part of this group, and thus, they adhered to the programme sessions.
*“I liked being in this group because I got to learn something new. Because I would never have practiced Yoga any other way, but when I was recruited to this group (intervention group), I thought I would learn something new. I thought of trying it because I love to do a new thing, so I found this thing right”.*(Age: 47 years; Male; Intervention)

**Good venue and other support provided for programme sessions**: Some participants mentioned specific factors because of which they could attend the programme sessions. For example, proximity to the programme venue helped them participate in the sessions.
*“Yes, because I had time and also because this place was closer to my house. I had various reasons to come here. First, it is beneficial for our health, secondly because of time, thirdly because it is close to my house. It was a combination of all three factors which helped me to participate in the study”.*(Age: 44 years; Male; Intervention)*“I would have skipped it if it was far from my home”.*(Age: 35 years; Female; Intervention)

Participants also discussed their experiences at the venue, which influenced adherence to the programme sessions. Many participants felt that the venue was comfortable, and the hall was big. They felt comfortable at the venue while practising Yoga poses.
*“No, it (venue) was comfortable only. The group was small. It was comfortable, there was no problem in that (venue)”.*(Age: 36 years; Male; Intervention)

Some participants reported that support and props provided as part of this feasibility trial, such as mats, pen drive and travel reimbursement, motivated them to attend the programme sessions.
*“…nobody in this world will give you anything for free, including Yoga classes, CDs, Yoga Mats, uniform etc., and nobody is bothered about you. Everybody thinks about themselves only. If the government (Research Team) is giving us Yoga classes, uniforms, Yoga mats for free, why should not we go there?”*(Age; 33 years; Male; Intervention)

**Professional behaviour of the YOGA-DP instructors**: Invariably, at both sites, all the intervention group participants had a good experience with the YOGA-DP instructor and received all the help they required at the site. Some other participants felt gratitude for the YOGA-DP instructor as they were not charging for the sessions and imparting training for the betterment of the participants’ health. They also mentioned that they acquired motivation from the YOGA-DP instructor to come for the sessions.
*“They used to call us frequently. YOGA-DP instructor was never strict with us, even if we had to miss a few sessions due to some personal reasons. He would motivate us for the next session. He would explain everything nicely about the exercise and body. He was not benefiting from that personally, but he was doing that for us as he never charged money. He was doing everything for us. He did not think about his benefits but thought about us”.*(Age: 36 years; Male; Intervention)

Some participants mentioned that they liked the YOGA-DP instructor’s approach of demonstrating all the Yoga poses by practising in front of participants. They also appreciated the YOGA-DP instructor for their problem-solving attitude. The demonstration of Yoga poses by the YOGA-DP instructor helped participants in practising them with precision.
*“He used to work hard with me. I have seen trainers at other places who teach you Yoga once or twice and after that ask you to do it on your own. But he (YOGA-DP Instructor) stayed with me throughout and used to practice Yoga with me. Since I did not do Yoga before, so he would teach me by practising on his own”.*(Age: 43 years; Male; Intervention)

Some participants pointed out that the YOGA-DP instructor was punctual, and they never had to wait at the site. They said YOGA-DP instructors provided care to them and used to ask them about their well-being and Yoga practice at home, which made them adhere to the programme sessions at the site.
*“Because if we had to come here by 6 am, YOGA-DP instructor would be here at that time, even before that. We never had to wait for her. She fully cooperated with us”.*(Age: 44 years; Female; Intervention)
*“If they did not call us (for supervised sessions), they used to ask about it (unsupervised sessions at home) when we used to go for sessions site-based (supervised) sessions. They used to ask about our sessions, if we had filled our notebook (YOGA-DP Diary), about the duration of our session, diet control etc”.*(Age: 44 years; Female; Intervention)

**The new learning experience of Yoga**: One participant was motivated to participate in the feasibility trial because she found this research good. She wanted to step out and learn something new, which motivated her to participate. Innovative experience and perceived lack of harm because of Yoga positively influenced participants’ decisions to participate in the intervention group.
*“One reason was that when we step outside, we learn something or the other. Another reason was if Yoga does not benefit us, it would not harm us either. So, I can take out 1 h for Yoga, and there is no harm in doing it”.*(Age: 35 years; Female; Intervention)

**Improvements to physical and mental well-being**: Positive physical and mental changes after joining the programme session were mentioned, such as changes in body shape, weight and flexibility. Some participants mentioned that their stress level had reduced and they could sleep appropriately, which motivated them to practice Yoga poses.
*“Yes, there has been a lot of changes physically. My belly size had increased way too much, which is fine now. I feel a lot more active than before”.*(Age: 40 years; Female; Intervention)

Some of them also witnessed a change in their behaviour and lifestyle after participating in the programme sessions. Yoga poses also helped them with managing their stress and anxiety, which motivated them to come for the programme sessions.
*“Earlier, I used to get irritated and got angry a lot. After these sessions, I feel like staying cool. If my kids are doing something, I am like, “It is okay, and you can continue doing it for a while”. I feel as if some positivity is there after doing this (practicing Yoga)”.*(Age: 44 years; Female; Intervention)

**Scheduling programme sessions as per participants’ preferences**: Some participants reported developing a routine because of participating in this intervention. They mentioned they started getting up early in the morning and practising Yoga more systematically in a time-bound fashion. Yoga helped many participants follow a disciplined lifestyle which motivated them to adhere to the programme sessions.
*“Earlier I used to get up late but now I get up by 6 or 7 am. After that I practice Yoga. It is better than what it was before”.*(Age: 36 years; Male; Intervention)*“Good things have happened because previously I used to practice whenever I got time, and now madam (YOGA-DP Instructor) started teaching yoga systematically and properly, and I am following that”.*(Age: 33 years; Female; Intervention)

Some participants felt excited to attend the programme sessions, and they also asked the YOGA-DP instructor for extra sessions. One participant felt trapped after getting recruited to the intervention group; however, he started liking the programme sessions with time and the help of the YOGA-DP instructor. Thus, participants’ interest in Yoga also motivated them to participate in the programme sessions infallibly.
*“Yes, in the beginning, I felt like I got trapped. I am telling you. But our YOGA-DP instructor said that “try it once or twice; after that, you share your experience”. When I started, I felt happy. I thought that this was a good thing, and gradually I developed an interest in this. It was better than that because I do not like restrictions”.*(Age: 47 years; Male; Intervention)

Some participants also had a specific liking for different Yoga poses. These Yoga poses helped them start their day with positive energy. Liking for various Yoga poses also made many participants adhere to the programme sessions.
*“I liked a few things in the starting, one was sun salutation and boat pose also, I had to raise my leg and sit. Everything was new to me”.*(Age: 47 years; Male; Intervention)
*“I liked doing sun salutation, the whole cycle of it. I used to sweat in it and feel energy afterwards. I liked it. I like doing it at home too, even if I am able to do only one cycle. After waking up in the morning, I do one cycle of it, get activated and get on with my work”.*(Age: 35 years; Female; Intervention)

Some participants joined the programme sessions as they were free, while others joined because they felt grateful that someone else was making efforts to take care of their health. The free viability of Yoga training and the selfless attitude of the YOGA-DP instructors facilitated participation in the intervention group.
*“…what happens is, people, pay to do Yoga outside (instructors or Yoga institutes) and here we were taught Yoga for free, so I felt nice about it”.*(Age: 57 years; Male; Intervention)

While discussing session preferences, many participants preferred sessions on weekends as they were free from family and professional responsibilities. Many also preferred morning sessions. Thus, the availability of sessions as per participants’ convenience helped participants to come for the programme sessions.
*“My kids have their holidays on weekends, so we used to go to classes while our kids were asleep. Also, our main objective was to do Yoga on an empty stomach in the morning; that is why we used to come in the morning”.*(Age: 44 years; Female; Intervention)
*“On Sundays, kids have their holiday, so a person is mostly free during morning and evening. On Saturdays too, I am busy as I drop my kids to school”.*(Age: 43 years; Male; Intervention)

Most participants felt the frequency of the programme sessions should be increased from two days as they wanted to come for the sessions more frequently. Some participants also requested the study team to continue the sessions even after the completion of the feasibility trial.
*“I did not find anything lacking in these sessions; everything was perfect. All I can say is that you can perhaps increase the number of classes from two (sessions) a week. There are seven days, so out of that (sessions) can be organised on three days perhaps”.*(Age: 35 years; Female; Intervention)

Participants also preferred group sessions over individual sessions. They believed individual sessions were boring, whereas, in a group session, they learnt from each other, and in a group, they felt competitive and tried to outperform other participants. Thus, the participants who preferred group sessions were invited in a group which made them adhere to the programme sessions.
*“Group sessions are good, but individual session is not good. Individual sessions are boring. What will you teach a single person? Group sessions are more enjoyable than individual sessions. In a group, you get to learn by seeing others”.*(Age: 36 years; Male; Intervention)
*“The important thing is that it is good to practice Yoga in a group, as we compete with each other. As I am doing better than him, he is not doing better than me”.*(Age: 33 years; Male; Intervention)

## 4. Discussion

This qualitative study explored trial-and intervention-related barriers and facilitators within a feasibility RCT of a Yoga-based intervention to prevent T2DM among high-risk people in India. The various trial-related barriers included inadequate information about the recruitment and randomisation processes, as explored in previous studies [27,28]. One of the possible reasons for inadequate information among participants could be the gap (six months) between the trial and in-depth interviews, due to which they could not recall the procedure. Many participants in the control group felt that they were recruited into this group because of their blood glucose level as compared to the participants in the intervention group. Some of them knew about the two groups in the feasibility trial but were not concerned about how and why they were recruited to a certain group and were not aware of the aim of the randomisation. Some of them also had poor experiences regarding the enhanced care leaflet. Most of the participants in the trial were happy with the blood sample collection and appreciated that the site made appropriate arrangements for the participants according to their convenience. However, most suggested that a blood test should have happened after three months to provide information about their health. However, some participants accepted the frequency of the blood tests and felt that a test before six months would not have shown anything about their health.

We systematically developed the YOGA-DP intervention after reviewing the scientific literature and summarised “the heterogeneous contents of successful and relevant Yoga interventions” [15]. We involved a range of stakeholders, including healthcare, medical and Yoga experts and practitioners and the public, to explore the issues of safety and acceptability [15]. Some of the barriers reported by intervention group participants included difficulty in comprehending the language of the programme booklets, difficulty in capturing duration and sequence of the Yoga poses in the programme diary, poor quality Yoga mats, problems with the Yoga videos, difficulty in practising Yoga poses and household commitment during the unsupervised sessions. A previous study on barriers and facilitators in physical activity participation among women also identified that lack of time prevented many from getting involved in physical activity [29]. We also found that group sessions could be a facilitator (for those who found them motivating) or a barrier (for those who preferred individual sessions or were concerned with gender or mixed-age groups). A previous study has also highlighted that fear of being compared to others in the Yoga session was a barrier to participation in the study [30].

There were many facilitators, such as adequate information provided on T2DM prevention and self-care, good venue and other support provided for the programme sessions, cordial behaviour of the YOGA-DP instructors, improvements to physical and mental well-being, learning something new and scheduling programme sessions as per participants’ preferences. The desire to improve physical health and mood and try something new were the few facilitators to participation in a previous study on Yoga intervention as well [30].

Reasons to take part included the prevention of T2DM and weight loss. Other participants referred to the YOGA-DP instructor’s altruism and the research staff that motivated them to adhere to the programme sessions. Previous studies have also highlighted that conditional altruism made many participants participate in a study [31,32]. For other participants, free Yoga classes and proximity to the venue was essential in keeping them motivated to adhere to the programme sessions.

Most participants reported no challenges in attending the programme sessions and practising the exercises, and no serious adverse events were recorded. Muscle pain and soreness were reported by some at the start of the session, which they overcame with their regular practice. This is consistent with prior work on reporting similar symptoms among Yoga participants [33].

Intervention group participants found the programme booklet helpful while practising at home as compared to the video, which many of them could not watch. However, almost all participants reported that they could not practice Yoga at home in a disciplined manner, following recommended frequency and duration, because of the unsupervised nature of the home-based sessions. They did not have any motivation to adhere to the programme session at home as there was no one to help them with their practice and rectify their mistakes.

Intervention group participants also mentioned that they had developed a very good relationship with the YOGA-DP instructors because of their positive and professional attitude. In Bengaluru, an opposite-sex YOGA-DP instructor was acceptable, although this was not the case at the Delhi site. While developing our intervention, we had already taken the gender of participants and YOGA-DP instructors into consideration and recruited both male and female YOGA-DP instructors for the participants to make the intervention more acceptable [15]; however same-sex YOGA-DP instructors were not available on some occasions.

The crucial facilitators based on the positive experience of participants included free blood tests, cordial behaviour of the site staff and improvements to physical and mental well-being. A number of suggestions were made for the future main RCT, including improving the quality of the Yoga mats, readability and understanding of the YOGA-DP booklets and the programme video (and/or making it available online or via WhatsApp) and increasing the frequency of YOGA-DP sessions.

### Strengths and Limitations of the Study

To the best of our knowledge, this is the first qualitative study identifying and exploring the barriers and facilitators in conducting a Yoga-based trial to prevent T2DM among high-risk individuals in India. One of the strengths of this study is that we were able to recruit participants from diverse sociodemographic backgrounds, age-range and languages spoken, which helped in identifying and exploring the intervention- and trial-related barriers and facilitators in a more inclusive way. Though we started reaching data saturation in the responses of participants after six or seven interviews in both groups, we continued interviewing more participants to capture any unique specific information [19,20].

One of the limitations of this study was that many interviews were conducted over the telephone due to the COVID-19 pandemic, which hampered ice-breaking with the participants and observation of their body language and facial interaction while conducting the semi-structured interview. Though the researcher spoke to the participants twice or thrice before the actual interview, in-person interviews could have generated more data based on participant observation as some of the telephonic interviews conducted in Kannada and Hindi were short in duration. Another limitation was that participants who were lost to follow-up when contacted over the phone [34], did not agree to participate in this qualitative study. One intervention group participant who stopped the intervention but continued participating in the trial was also interviewed.

## 5. Conclusions

We identified and explored participants’ trial- and intervention-related barriers and facilitators, which will inform the design of the planned definitive RCT design and intervention, as well as other Yoga interventions and RCTs. We will address the issues that negatively influenced the trial and intervention participation and adherence and promote the facilitators to improve trial and intervention participation and adherence. In this study, we could identify an almost equal number of barriers (*n* = 12) and facilitators (*n* = 13); however, intervention-related barriers and facilitators were more than for participation in the trial.

## Figures and Tables

**Table 1 ijerph-19-05514-t001:** Sociodemographic details of participants.

		*n* = 25 Intervention = 13 Control = 12	Intervention Group	Control Group
Sociodemographic characteristics				
Age (years)		25–64 (range) (Median 41)	25–60 (range) (Median 41.5)	28–64 (range) (Median 41)
Sex	Female	13	7	6
	Male	12	7	5
Marital status	Married	25	14	11
Formal education (years)	≤10	4	1	3
	>10	21	13	8
Employed	Yes	15	7	8
	No	10	7	3
Gross monthly household income (INR)		10,000–245,000 (range) (median 25,000)	10,000–80,000 (range) (median 20,000)	12,000–245,000 (range) (median 42,000)
Family history of diabetes		11	6	5

**Table 2 ijerph-19-05514-t002:** Themes and sub-themes.

Themes	Sub-Themes
**Barriers to trial participation**	Detailed information about recruitment and randomisation processes
	Poor experience in the control group regarding the enhanced care leaflet
	The negative influence of non-participants
	Frequency of the blood test (e.g., FBG)
**Facilitators to trial participation**	Adequate information about the trial and related processes
	Free blood tests and positive experience of the testing process
	To gain adequate information to prevent T2DM
	Professional behaviour of the site staff
	The positive influence of friends
	Trust in the trial sites and the range of healthcare services they provide
**Barriers to participation in the intervention**	Difficulty in reading and understanding the language of the programme booklets
	Difficulty in capturing duration and sequence of the Yoga poses (asanas) in the programme diary
	Difficulty in using the programme video during unsupervised sessions
	Poor quality of Yoga mats
	Household commitment and unavailability of the YOGA-DP instructors hindered unsupervised programme sessions
	Missed supervised sessions due to unplanned travel
	Difficulty in practising Yoga poses
	Hesitation in attending programme sessions with the YOGA-DP instructor of the opposite sex
	Hesitation in attending group programme sessions
**Facilitators to participate in the intervention**	Programme booklets helped in adhering to the unsupervised programme sessions
	Adequate information was provided on T2DM prevention and self-care
	Good venue and other support provided for programme sessions
	Professional behaviour of the YOGA-DP instructors
	The new learning experience of Yoga
	Improvements to physical and mental well-being
	Scheduling programme sessions as per participant’s preferences

## Data Availability

The raw data supporting the conclusions of this article will be made available by the authors, without undue reservation.

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
