# Peer review of "Feasibility Trial of Yoga Programme for Type 2 Diabetes Prevention (YOGA-DP) among High-Risk People in India: A Qualitative Study to Explore Participants’ Trial- and Intervention-Related Barriers and Facilitators"

_ijerph, 2022, doi:10.3390/ijerph19095514_

Round 1

Reviewer 1 Report

Hi

Sport for health and the prescription of physical activity in chronic pathologies is a topical issue and we must encourage all work in this direction.

I have put some comments in the text.
I don't think that having the patients' comments written in the manuscript adds anything to the proposed study.
It is hard to see what the authors are going to do to avoid all the pitfalls found in this study and make it a larger scale study.
I did not see any comaprative results between the two groups even though the sample is small apart from citing all the difficulties encountered.

Author Response

We thank the reviewers for sharing their valuable feedback, and our responses to the individual points are listed below.

  1. do you think that it is possible to have s selected biais between the Yoga group and the control group because the control group as you say is only composed by people who refused to participate at this study?

Thank you for this comment. In line 116 we have mentioned that “control group participants who had completed six months of participation were recruited from both trial sites”. We made efforts to have an equal representation of both groups (13 intervention group, 12 control group) for this qualitative study to avoid selection bias.

This is a qualitative paper from a feasibility randomised control trial, and the primary outcome paper is under review in another journal. We randomly allocated intervention and control groups to the eligible participants.

  1. why don't you put up a flow chart explaining how you arrived at only 25 participants for this study?

Thank you for this comment. We have added table 1, representing the characteristics of the participants. Since this is a qualitative study, we followed the principle of saturation of data to recruit 25 participants, and we have mentioned the same in lines 790 and 791- “Though we started reaching data saturation in the responses of participants after six or seven interviews in both groups, we continued interviewing more participants to capture any unique specific information.”

  1. I'm not sure that the participants' comments do not add anything to the article and could be removed

Thank you for this comment. We have used quotes to support our research claims, exemplify ideas, share experiences, and present the emotions of the participants. This is the norm followed and required in qualitative studies to support the themes in the manuscript.

  1. What did you mean about lack of adequate information ?

Thank you for this comment. We have now replaced the word ‘inadequate’ with ‘detailed’ in lines 203, 217, 2018

  1. This too may induce a bias

Thank you for this comment, and we apologise for the lack of clarity in the sentence. We have amended the below sentence (Line 270, 271, 272):

“Some of the participants shared that they understood the randomisation and recruitment processes. They mentioned that the site staff had informed them about randomisation and recruitment to their respective groups, and they decided to be a part of that group and follow the processes.”

  1. Only one person's opinion!

Thank you for this comment. We have added one more quote to the paragraph.

“We are getting aware of our health and wellbeing. Whatever disease – can’t call it a disease, or (we can say) any problem related to our body that we (may) have. They are giving us critical information about it, that we can prevent that problem without taking any medication (by practising Yoga), then what can better than this?” (Age: 40 years; Female; Control)

  1. It will be preferable to put this table at the beginning of the results

Thank you for this comment. This is the discretion of the journal editorial board and the format followed by the journal.

  1. Where are the data on the glycaemic assessment of patients? Where is the comparison between the two groups on all these data?

Thank you for this comment. We have already done the glycaemic assessment of patients and comparison between the two groups - the randomised controlled trial manuscript is under review in another journal. In this qualitative study, we intend to understand participants’ experiences and perceptions of taking part in the Yoga-based feasibility Randomised Control Trial for the prevention of T2Dm.

  1. where are the data of patients satisfied with the yoga-DP programme

Thank you for this comment. We have tried to capture the positive experience of the participants through two major themes on trial and intervention-related facilitators, which also signify satisfaction among participants.

  1. Maybe put in conclusion the future study you want to set up and how you will fight against the reluctance of patients

Thank you for this comment. We have added a sentence in the conclusion as suggested-

“We will address the issues that negatively influenced the trial and intervention participation and adherence. Similarly, we will promote the facilitators to improve trial and intervention participation and adherence”.

Reviewer 2 Report

This is a nice story describing the problems a researcher will be confronted  when trying  to get participants for a trial and it is humorous to read that problems like apologizing for not showing up are the same all over the world, I recognize many remarks the participants made. That made the paper gave me a familiar feeling.

What I miss being a reader not knowledgeable with Yoga, what “ASANAS”is.  Furtheron In line 665/667 I have no idea what Surya Namaskar an Naukasian is???

What I also miss is what sort of blood test were planned to do.  That would have made more clear to a reader what sort of population you hope to include in the upcoming RCT. Maybe you can made that more clear.

I think it must be explained when speaking of a  population high risk for developing diabetes, why your population is at high risk.

But your purpose to tell others what the difficulties are when trying to start a trial is certainly a success with this paper. For that reason I am in favour that the paper will be published

Author Response

We thank the reviewers for sharing their valuable feedback, and our responses to the individual points are listed below.

  1. What I miss being a reader not knowledgeable with Yoga, what “ASANAS”is. Furtheron In line 665/667 I have no idea what Surya Namaskar an Naukasian is???

Thank you for this comment. Asana means Yoga Poses, which we have mentioned in line 365.

We have added the English name for Surya Namaskar (Sun Salutation) and Naukasana (Boat Pose) in lines 671, 672, 674. 

  1. What I also miss is what sort of blood test were planned to do. That would have made more clear to a reader what sort of population you hope to include in the upcoming RCT. Maybe you can made that more clear.

Thank you for this comment. We have now added the name of the blood tests (e.g., FBG) in lines 90-91.

  1. I think it must be explained when speaking of a population high risk for developing diabetes, why your population is at high risk.

Thank you for this comment. We have made changes in lines 90 and 91. We have also mentioned it in the introduction section in lines 55, 56 and 57.

“We used a multipronged approach, such as door-to-door campaigns, posters and screening camps to identify potential participants, i.e., adults at high risk of T2DM  as their fasting blood glucose (FBG) level was between 100 and 125 mg/dL [18].”

“Another 77 million people in the country are at high risk of developing type 2 diabetes mellitus (T2DM) because of raised blood glucose levels but below the established threshold for T2DM [2].”

Round 2

Reviewer 1 Report

Thank you for your answers and proposed corrections, but it is a pity to have dissociated this manuscript from the randomised controlled trial which is being reviewed in another journal, because we are still missing some of the data even though you have answered them.
I understood that it was a qualitative study.
Personally, I still think that there are too many patient comments that detract from the reading of the article.

Author Response

Dear Reviewer,

Thank you for your valuable feedback and comments, which helped strengthen our paper. As suggested, we have deleted a few quotes in the manuscript to make it readable. 

Thanks and regards,

Pallavi Mishra
